# Nanomedicine-Based Gene Delivery for a Truncated Tumor Suppressor RB94 Promotes Lung Cancer Immunity

**DOI:** 10.3390/cancers14205092

**Published:** 2022-10-18

**Authors:** Sang-Soo Kim, Caroline Doherty, Manish Moghe, Antonina Rait, Kathleen F. Pirollo, Joe B. Harford, Esther H. Chang

**Affiliations:** 1Department of Oncology, Lombardi Comprehensive Cancer Center, Georgetown University Medical Center, Washington, DC 20057, USA; 2SynerGene Therapeutics, Inc., Potomac, MD 20854, USA; 3College of Medicine and Science, Mayo Clinic, Rochester, MN 55905, USA

**Keywords:** non-small cell lung cancer, nanomedicine, retinoblastoma, tumor-targeted delivery

## Abstract

**Simple Summary:**

Advanced therapies have provided substantial clinical benefits in oncology, but low response rates in lung cancer patients remain challenging. Using preclinical models of human lung cancer, we investigated the anti-tumor potency of nanomedicine-based gene delivery for the tumor suppressor RB94 and explored the mechanisms underlying its activity. This nanomedicine (scL-RB94) elicited significant anti-tumor responses reflecting enhanced tumor immunogenicity and reduced immunosuppression. The potential of scL-RB94 to improve outcomes in lung cancer patients is discussed.

**Abstract:**

Because lung cancer remains the most common and lethal of cancers, novel therapeutic approaches are urgently needed. RB94 is a truncated form of retinoblastoma tumor suppressor protein with elevated anti-tumor efficacy. Our investigational nanomedicine (termed scL-RB94) is a tumor-targeted liposomal formulation of a plasmid containing the gene encoding RB94. In this research, we studied anti-tumor and immune modulation activities of scL-RB94 nanocomplex in preclinical models of human non-small cell lung cancer (NSCLC). Systemic treatment with scL-RB94 of mice bearing human NSCLC tumors significantly inhibited tumor growth by lowering proliferation and increasing apoptosis of tumor cells in vivo. scL-RB94 treatment also boosted anti-tumor immune responses by upregulating immune recognition molecules and recruiting innate immune cells such as natural killer (NK) cells. Antibody-mediated depletion of NK cells blunted the anti-tumor activity of scL-RB94, suggesting that NK cells were crucial for the observed anti-tumor activity in these xenograft models. Treatment with scL-RB94 also altered the polarization of tumor-associated macrophages by reducing immune-suppressive M2 macrophages to lower immune suppression in the tumor microenvironment. Collectively, our data suggest that the efficacy of scL-RB94 against NSCLC is due to an induction of tumor cell death as well as enhancement of innate anti-tumor immunity.

## 1. Introduction

With an estimated 1.8 million deaths globally in 2020, lung cancer remains the leading cause of cancer-associated death [1]. Non-small cell lung cancer (NSCLC) comprises 85% of lung cancer cases, and the majority of NSCLC patients are diagnosed at advanced stages, resulting in historically poor median overall survival (OS) and 5-year survival rates [2]. Platinum-based chemotherapies, which have been the first line of treatment for several decades, only modestly improved survival times but were accompanied by very high toxicity [3]. Molecularly targeted therapies such as epidermal growth factor receptor (EGFR), anaplastic lymphoma kinase (ALK), BRAF, and mitogen-activated protein kinase kinase (MEK) inhibitors have produced robust responses and prolonged progression-free survival. However, these are effective for only small subsets of lung cancer patients with tumors that harbor specific genetic alterations [4]. Unfortunately, drug resistance develops in these patients and nearly all who initially responded to treatments developed subsequent progressive disease [5]. More recently, immune checkpoint inhibitors have shown promising prolonged and durable responses [6]. However, agents acting via immune checkpoint blockade also benefit only a minority (10–20%) of patients with advanced or metastatic stages of NSCLC [7]. Thus, investigation of novel approaches to treat or to augment current therapies is warranted to improve outcomes for patients with NSCLC.

The retinoblastoma protein (RB) is a multifunctional tumor suppressor, with the RB pathway disrupted in a variety of human malignancies including retinoblastoma [8,9]. Studies have shown that adenovirus-mediated cancer gene therapy using wild-type RB inhibits tumor cell growth [10,11]. However, wild-type RB gene therapy has limited efficacy due to rapid phosphorylation and inactivation of ectopically expressed RB protein [12,13,14]. To circumvent the problem of the functional inactivation of RB protein, a truncated variant of RB lacking the 112 N-terminal amino acids was identified [11]. Compared to its full-length counterpart of approximately 110 kDa, this 94 kDa variant (RB94) has significantly elevated anti-tumor activity due to an extended half-life and an ability to remain in an active hypo-phosphorylated form [11,15]. RB94 protein can cause rapid telomere erosion and chromosomal fragmentation in tumor cells, leading to cytotoxicity irrespective of the tumor cells’ native RB status [10,11,16]. Remarkably, despite the potent anti-cancer cytotoxicity of RB94, normal human cells appear to have little or no susceptibility to RB94 toxicity [16,17,18].

To realize the therapeutic potential of RB94, we have developed a novel tumor-targeted nanomedicine for RB94 gene therapy, termed scL-RB94. In the scL-RB94 nanocomplex, a plasmid DNA encoding human RB94 is encapsulated within a cationic liposome whose surface is decorated with an anti-transferrin receptor (TfR) single-chain antibody fragment (TfRscFv) as a tumor-targeting moiety. TfRscFv selectively targets cancer cells by binding to the TfR that is overexpressed on the surface of tumor cells, and the nanocomplex is internalized via receptor-mediated endocytosis [15,19]. Our tumor-targeted TfRscFv/Liposome (termed scL for single chain Liposome) delivery system enables systemic delivery of gene therapy to cancer cells, while reducing exposure to non-cancerous tissues. A recent phase I human trial of scL-RB94 (known as SGT-94) as a single agent for patients with genitourinary cancers demonstrated a good safety profile and promising anti-tumor activities for some patients who experienced either partial or complete remission [19]. The remarkable tumor-specificity of scL-RB94 was demonstrated in this clinical trial, with RB94 detectible only in primary and metastatic lesions but not in adjacent normal tissue [19]. Although the safety and efficacy of this novel RB94 gene therapy agent was demonstrated in that clinical trial, the mechanisms underlying the observed anti-tumor effects have not yet been explained.

The urgent need for improved therapies for lung cancer prompted us to examine the effects of scL-RB94 against NSCLC. In this paper, we show that scL-RB94 is effective in treating mice bearing human NSCLCs by not only driving human tumor cell apoptosis but also boosting the host’s anti-tumor immune responses. Notably, the payload in the scL-RB94 nanocomplex is the human RB94 gene that suppresses growth of human tumor cells but is ineffective on mouse tumor cells, thereby necessitating investigation of RB94′s mechanism of action in human xenograft models [20]. Owing to the lack of T-cell-mediated immunity in xenograft models in athymic mice, we focused on innate anti-tumoral immunity. In two distinct xenograft models of human NSCLC (H292 and H358), we observed that scL-RB94 suppressed tumor proliferation and growth, triggered tumor cell death, induced immunogenic changes in tumors, and increased immune infiltration (e.g., NK cells) in the tumor microenvironment. Importantly, antibody-mediated depletion of NK cells resulted in a significant decrease of the tumor-suppressive effect of scL-RB94, indicating the contribution of NK cells to the anti-tumor activity of RB94 therapy in vivo. We also observed a decrease in immune-suppressive M2 macrophages that would alleviate immune suppression in the tumor microenvironment. Collectively, our data suggest that RB94 gene therapy via scL-RB94 could be a novel therapeutic option. The study findings provide a rationale for evaluation of SGT-94 in patients with advanced NSCLC.

## 2. Materials and Methods

### 2.1. Cell Lines

A549, NCI-H292, NCI-H596, and NCI-H358 were obtained from the Tissue Culture Core at the Lombardi Cancer Center. NCI-H23 was purchased from the American Type Culture Collection (Manassas, VA, USA). Cells were maintained at 37 °C in 5% CO_2_ in modified RPMI-1640 (H292, H596, H358, and H23, Corning Cellgro, Corning, NY, USA) or DMEM (A549, Corning Cellgro) supplemented with 10% fetal bovine serum (Sigma, St. Louis, MO, USA). Cell lines H292 and H596 were also supplemented with 1% sodium pyruvate, 1% 4-(2-hydroxyethyl)-1-piperazineethanesulfonic acid (HEPES), and/or 1% glucose (all Thermo Fisher, Waltham, MA, USA). All experiments were performed with mycoplasma-free cells. The status (wild type, mutant, deleted, and up/down regulation of transcription) of 5 genes significant in NSCLC tumorigenesis and present in the largest populations of NSCLC patients were determined for the studied human NSCLC cell lines using the Sanger Institute Catalogue Of Somatic Mutations In Cancer (COSMIC) database (Appendix A).

### 2.2. Nanocomplex Preparation

Cationic liposomes, referred to as Lip, consisting of 1,2-dioleoyl-3-trimethylammonium propane and dioleolylphosphatidyl ethanolamine (Avanti Polar Lipids, Alabaster, AL, USA) were prepared as previously described [15]. Plasmid DNA carrying the human RB94 gene was encapsulated in TfRscFv/Lip (termed scL for single-chain liposome) nanocomplex (scL-RB94, a.k.a., SGT-94) [15]. The sizes of the complexes were measured by dynamic light scattering using a Zetasizer (Malvern Panalytical, Malvern, UK). For experiments involving cell cultures, the complex was further diluted with serum-free medium. For animal injections, 5% dextrose was added to the nanocomplex preparation.

### 2.3. Measurement of RB94 Expression In Vitro

To determine the expression level of exogenous RB94 protein, Western blot analysis was performed using H358 cells (1.0 *×* 10^6^ cells/10 cm cell culture dish) after transfection with scL-RB94 (0.8 μg DNA/dish). Forty-eight hours post-transfection, the cells were collected and lysed in cold radioimmunoprecipitation assay (RIPA) buffer with protease inhibitors. The protein concentration was measured and 10 μg of total protein was separated on 4–12% Bis-Tris Midi gel (Invitrogen, Waltham, MA, USA), transferred to a nitrocellulose membrane, and hybridized with antibodies against RB (#3107, 1:10,000 dilution, QED Bioscience, San Diego, CA, USA) followed by incubation in a horseradish peroxidase-conjugated anti-mouse IgG (#31430, 1:5000 dilution, Thermo Fisher). Antibodies recognizing lamin B1 (#sc-377000, 1:1000 dilution, Santa Cruz Biotechnology, Dallas, TX, USA) were utilized as an internal control for protein loading. Chemiluminescent detection was carried out using SuperSignal West Dura Extended Duration Substrate (Thermo Fisher). Quantification of protein bands was carried out using ImageJ software (https://imagej.nih.gov/ij, accessed on 1 May 2019).

### 2.4. Cell Viability Assay

Cell viability was measured by sodium 3′-[1-(phenylamino-carbonyl)-3, 4-tetrazolium]-bis(4-methoxy-6-nitro)-benzenesulfonate (XTT) assay (Polysciences, Warrington, PA, USA). Human NSCLC cells (2.0 × 10^3^ cells/well in 96-well plates in triplicate) were treated with either the scL-RB94 or the scL nanocomplex without payload at various concentrations, as previously described [15]. Cell viability was determined 72 h post-treatment by XTT assay, and the IC_50_ values, i.e., the drug concentrations resulting in 50% cell death, were interpolated from the graph of the DNA concentration versus the fraction of surviving cells, using SigmaPlot 11.2 (Systat Softwate Inc., San Jose, CA, USA).

### 2.5. Caltericulin Assay

H358 cells were transfected with scL-RB94 at IC_50_ (4.1 µg DNA/dish). At 72 h post-transfection, the cells were harvested and stained with 10 μg/mL anti-CRT antibody (NBP-47518AF405, Novus Biologicals, Centennial, CO, USA) in phosphate-buffered saline containing 1% bovine serum albumin, on ice for 30 min. Cells were analyzed using an LSRFortessa flow cytometer (BD Biosciences, San Jose, CA, USA).

### 2.6. ATP Assay

H358 and H292 cells were transfected with scL-RB94 at IC_50_ (4.1 and 2.7 µg DNA/dish, respectively). At 72 h post-transfection, the cell culture supernatant was harvested. Levels of extracellular ATP in the cell culture medium were measured by luciferin-based ENLITEN ATP Assay (Promega, Madison, WI, USA) following the manufacturer’s instructions.

### 2.7. Animal Models

All animal experiments were performed in accordance with approved Georgetown University GUACUC protocols. For the xenograft tumor models, 5–6-week-old female athymic nude mice (Hsd:Athymic Nude-Foxn1^nu^, Envigo, Indianapolis, IN, USA) were inoculated subcutaneously on the lower back, above the tail, with either H358 or H292 cells (1 × 10^6^ or 1.5 × 10^6^ cells/site, respectively) suspended in Matrigel (Corning). Tumors were allowed to grow until they averaged 100–200 mm^3^ and mice were randomized into treatment groups. Mice with established tumors were systemically injected in the tail vein twice weekly either with nanocomplex delivering the control plasmid vector without RB94 (scL-vec) or with scL-RB94 (30 µg DNA/injection/mouse). In studies involving depletion of NK cells, 200 µg of anti-NK1.1 antibody (clone PK136, BioXCell, Lebanon, NH, USA) was given intraperitoneally 24 h prior to the injection of scL-RB94.

### 2.8. Immunohistochemistry

On day 5, after completing 2 treatments, tumors were harvested, fixed in formalin and embedded in paraffin block for immunohistochemistry (IHC). Tumor sections were stained using antibodies for Ki-67 (Agilent, Santa Clara, CA, USA), high mobility group box 1 (HMGB1, Sigma), Arg1 (Thermo Fisher), or iNOS (Thermo Fisher). Tumor sections were also subjected to terminal deoxynucleotidyl transferase dUTP nick end labeling (TUNEL) assay using an ApopTag Peroxidase In Situ Apoptosis Detection Kit (Millipore, Burlington, MA, USA). Images were captured using an Olympus DP70 camera on an Olympus BX61 microscope. Captured images were analyzed using the IHC Profiler plugin in ImageJ. At least 8 representative images from 4 tumors were utilized to quantify the staining results.

### 2.9. Flow Cytometry Analysis

A single cell suspension was prepared using a Tumor Dissociation Kit and gentleMACS Octo Dissociator with Heaters (Miltenyi Biotec, Bergisch Gladbach, Germany) according to the manufacture’s protocol. Cells were pre-labelled for viability with Zombie-NIR viability dye (BioLegend, San Diego, CA, USA) and stained where indicated with antibodies against HLA-ABC (311436), CD31 (102424), CD45 (103114 or 103151), CD11b (101228), F4/80 (123110), CD206 (141717), CD86 (105067), NK1.1 (108736) (all BioLegend), UL16 binding protein 1 (ULBP-1, FAB1380G), or ULBP-2 (FAB1298C) (all Novus Biologicals). Cells were analyzed using a LSRFortessa flow cytometer.

### 2.10. Western Blot Analysis

To determine the protein expression level, Western blot analysis was performed. For the in vivo study, two models of NSCLC xenografts (H358 and H292) were used. Mice with established tumors received two injections of scL-RB94 (30 µg/mouse/injection) over a 72 h period via the tail vein. At 48 h after the second injection, the mice were humanely euthanized, and the tumors harvested. The protein samples were isolated in cold RIPA buffer with protease inhibitors. The protein concentration was measured and 10 μg of total protein was separated on 4–12% Bis-Tris Midi gel (Invitrogen), transferred to a nitrocellulose membrane, and hybridized with antibodies against RB (#3107, 1:10,000 dilution), cleaved caspase-3 (cCasp3, #9661, 1:1000 dilution, Cell Signaling, Danvers, MA, USA), and cleaved poly (ADP-ribose) polymerase (cPARP, #5625S, 1:1000 dilution, Cell Signaling), followed by incubation in horseradish peroxidase-conjugated anti-mouse IgG (#31430, 1:5000 dilution) or anti-rabbit IgG (#7074S, 1:10,000 dilution, Cell Signaling). Antibodies recognizing either glyceraldehyde 3-phosphate dehydrogenase (GAPDH, #2275-PC-100, 1:1000 dilution, Trevigen, Gaithersburg, MD, USA) or lamin B1 (#sc-377000, 1:1000 dilution) were utilized as internal controls for protein loading. Chemiluminescent detection was carried out using SuperSignal West Dura Extended Duration Substrate. Quantification of protein bands was carried out using ImageJ software.

### 2.11. Quantitative RT-PCR

Total RNA was extracted using a PureLink RNA mini kit (Ambion, Austin, TX, USA) and reverse transcribed with an iScript cDNA synthesis kit (Bio-Rad, Hercules, CA, USA) following the manufacturers’ protocols. PCR was performed in triplicate using iTaq Universal Probes Supermix (Bio-Rad) and TaqMan gene expression assays (Life Technologies, Carlsbad, CA, USA) for human GAPDH (Hs02786624_g1), human interferon alpha-1 (IFNA1, Hs00256882_s1), human IFNA2 (Hs00265051_s1), human IFNB1 (Hs01077958_s1), human IL-15 (Hs01003716_m1), mouse IL-15 (Mm00434210_m1), human ULBP2 (Hs00607609_mH), human MHC class I polypeptide–related sequence A (MICA, Hs00792195_m1), and human MHC class I polypeptide-related sequence B (MICB, Hs00792952_m1). PrimePCR Probe assays (Bio-Rad) were used for human leukocyte antigen (HLA)-A (qHsaCEP0040111), human transporter associated with antigen processing 1 (TAP1, qHsaCEP0039851), and human TAP2 (qHsaCEP0040018). The assays were run on the StepOnePlus RT-PCR system (Life Technologies) and the relative mRNA expression was analyzed using StepOne software v2.3. via the ΔΔCt method with normalization to the corresponding sample’s value for GAPDH.

### 2.12. Transcriptome Analysis

Gene expression was measured on a commercially available gene panel (PanCancer Human IO 360) containing a total of 770 unique genes (NanoString Technologies, Seattle, WA, USA). Raw data were normalized using nSolver 4.0 software (NanoString Technologies), based on the geometric mean of the negative controls, internal housekeeping genes, and positive controls. Normalized counts from genes were log2-transformed and used for further analysis. Genes that were changed up or down by at least 50% with false discovery rate (FDR)-adjusted *p* < 0.05 were considered differentially expressed. We performed gene ontology (GO) analyses using the ToppGene Suite online tool (https://toppgene.cchmc.org/, accessed on 9 June 2022).

### 2.13. Statistical Analysis

Presented data represent mean ± standard deviation (SD). Statistical significance was determined by Student’s *t* test or one-way analysis of variance (ANOVA) followed by a Bonferroni-corrected Student’s *t* test, to compare data between two or more groups. For all comparisons, *p* < 0.05 was considered significant. A log-rank test was used for survival studies. All graphs were prepared and statistical analysis carried out using SigmaPlot 11.2 (Systat Software).

## 3. Results

### 3.1. Size Determination of scL-RB94 Nanocomplex

The size of the tumor-targeted liposome nanocomplexes was analyzed by dynamic light scattering on a Malvern Zetasizer. The liposome nanocomplex without payload (scL) measured approximately 28.56 ± 1.89 nm (number values) with a zeta potential of 56.09 ± 1.63 mV (Appendix A). The scL-RB94 nanocomplex was approximately 96.41 ± 23.39 nm with a zeta potential of 27.49 ± 0.76 mV, while scL-vec was approximately 92.25 ± 26.74 nm with a zeta potential of 31.35 ± 2.27 mV, demonstrating that these complexes were in the nanosize range. The polydispersity of the complex size distribution was found to be within the recommended range (less than 0.3) for all of the complexes, indicating a homogenous population of liposome nanocomplexes [21]. The positive zeta potential of the complexes was optimal for interactions with cell membranes.

### 3.2. scL-RB94 Effectively Kills Human NSCLC Cells and Suppresses Tumor Growth

First, we confirmed the expression of exogenous RB94 protein for in vitro and in vivo samples treated with scL-RB94 nanocomplex. Western blot analysis showed distinct exogenous RB94 protein bands at about 94 kDa, along with endogenous full-length RB protein at about 110 kDa (RB110) in H358 NSCLC cells and tumor tissues (Figure 1A). However, untreated controls only showed endogenous full-length RB protein. To assess anti-tumor activity of exogenous RB94 protein, five human NSCLC cell lines (H358, H292, A549, H596, and H23) were transfected with either scL nanocomplex or scL-RB94 nanocomplex, and XTT assay was performed at 72 h to determine tumor cell survival. While scL displayed no tumor cell killing effect with the equivalent amount of nanocomplex, the scL-RB94 showed dose-dependent tumor cell killing with IC_50_ of 24.1 ± 0.2 ng and 15.7 ± 0.5 ng of plasmid DNA per well in H358 and H292 cells, respectively (Figure 1B and Appendix A), suggesting that scL-RB94-mediated exogenous RB94 protein expression was able to induce tumor cell death. All five of the tested human NSCLC cell lines demonstrated tumor cell death with scL-RB94 treatment, despite marked histopathological and genetic differences (e.g., RB status) in these tumors (Appendix A, Appendix A).

To study the effect of scL-RB94 treatment against human NSCLC in vivo, mice bearing subcutaneously established xenografts of either H358 or H292 were treated with scL-RB94 via tail-vein injections (30 µg of plasmid DNA/injection). There was significant inhibition of tumor growth after treatment with scL-RB94 compared with untreated mice (Figure 1C and Appendix A). No significant inhibition of tumor growth was seen with tumor-targeting nanocomplex delivering the control plasmid vector without RB94 (scL-vec), while scL-RB94 treatment showed a significant retardation of H358 tumor growth until day 21 compared with the untreated control (Figure 1C). At day 21, the average volume of H358 tumors from mice that received scL-RB94 treatment was not significantly different from the initial volume (i.e., 2 days prior to initiation of treatment), whereas the volume of tumors from mice either untreated or treated with scL-vec increased approximately three-fold from their initial volume (Figure 1C). Weights of the H358 tumors in mice treated with scL-RB94 were significantly lower than those in untreated mice (Figure 1D). Similar inhibition of tumor growth was observed in a repeated experiment with H358 tumors (Appendix A), confirming the anti-tumor effect of scL-RB94 treatment. These results indicate that the observed anti-tumor activity was specific to the expression of exogenous RB94 rather than a result of the delivery system itself or merely the introduction of exogenous DNA. Similarly, scL-RB94 treatment of mice bearing H292 tumor xenografts revealed comparable results to those observed with H358 tumors (Appendix A). H292 tumors from untreated mice weighed approximately four times more than those from mice treated with scL-RB94 (Appendix A). The bodyweight of scL-RB94 treated mice remained stable, indicating few, if any, systemic side effects (Appendix A). Notably, scL-RB94 was also found to have a good safety profile according to our toxicology study using non-tumor bearing BALB/c mice. This reflects the good safety profile observed in the SGT-94 clinical trial [19].

To further understand the effects of scL-RB94 on NSCLC, we assessed the proliferation and death of tumor cells in H358 and H292 tumors, using immunohistochemical (IHC) analysis. IHC analysis confirmed a significant decrease in positive staining for Ki-67, indicative of tumor cell proliferation, in H358 and H292 tumors treated with scL-RB94 compared with untreated tumors (Figure 1E, upper panels). In addition, there was a significant increase of positive TUNEL staining, indicative of DNA fragmentation and a hallmark of apoptosis, in both tumor models treated with scL-RB94 compared with untreated tumors (Figure 1E, lower panels). Western blot analysis of these tumors further revealed a significant expression of cleaved caspase-3 (cCasp3) and cleaved PARP (cPARP) in tumors from scL-RB94-treated mice, but not those from the untreated animals, suggesting that caspase-dependent apoptosis was occurring only in the tumors receiving scL-RB94 treatment (Figure 1F). These data together indicate that expression of exogenous RB94 protein by scL-RB94 nanocomplex effectively suppresses tumor growth by modulating proliferation and apoptosis of NSCLC cells.

### 3.3. scL-RB94 Increases Immunogenicity of NSCLC Cells

To investigate whether the scL-RB94 treatment could activate anti-tumor immune responses, we analyzed markers involved in immunogenic cell death (ICD), a type of cancer cell death eliciting an immune response. H358 cells in culture were transfected with either empty nanocomplex without payload (scL), nanocomplex carrying control plasmid vector without RB94 (scL-vec), or nanocomplex carrying RB94 plasmid (scL-RB94). FACS analysis of cells treated with scL-RB94 revealed a significantly increased surface expression of calreticulin (CRT), which is a hallmark of ICD, relative to untreated cells and controls treated with either scL or scL-vec (Figure 2A). In addition, extracellular release of ATP, a damage-associated molecular pattern (DAMP) that serves as innate immune ligands, was also significantly increased in the culture supernatant from cells treated with scL-RB94 compared with those from control groups (Figure 2B). Supplementing these in vitro results, there was a significant increase of HMGB1 expression in H358 and H292 tumors treated with scL-RB94 (Figure 2C), demonstrated by IHC, a method previously used for testing ICD in patient biopsy specimens [22]. Together, these data indicate that expression of exogenous RB94 is responsible for induction of ICD and immunogenic changes of NSCLC cells.

In addition, RT-PCR analysis revealed that scL-RB94 treatment could significantly increase the transcription of type I interferons (IFNa1, IFNa2, and IFNb1) in H358 cells in culture, compared with untreated cells (Figure 2D). Interestingly, although scL treatment caused no increase, treatment with scL-vec nanocomplex was able to increase type I IFNs, indicating a potential involvement of the STING pathway by introducing exogenous plasmid DNA. However, scL-RB94 treatment was able to further significantly increase type I IFNs compared with scL-vec, suggesting RB94 modulated production of IFNs beyond STING activation. To investigate the effects of scL-RB94 treatment on antigen processing and presentation, we evaluated the human major histocompatibility complex (MHC) molecule HLA-A and antigen transporters TAP1 and TAP2 using quantitative RT-PCR. Transfection with scL-RB94 significantly increased the transcription of HLA-A, TAP1, and TAP2 in a time-dependent manner in both H358 and H292 cells, compared with untreated cells (Figure 2E and Appendix A). Similar upregulation of HLA-A, TAP1, and TAP2 was also observed in vivo with scL-RB94 treatment of H358 tumors (Figure 2F). FACS analysis further confirmed increased expression of human MHC class I molecules HLA-ABC on the surface of H358 tumor cells in vivo after scL-RB94 treatments, compared with untreated tumors (Figure 2G). Taken together, these results suggest that scL-RB94 treatment generally increases the overall immunogenicity of tumors by upregulating immune-recognition molecules related to innate and adaptive immune systems.

### 3.4. scL-RB94 Enhances Host Immune Responses

We further investigated whether the scL-RB94-mediated enhancement of tumor response was associated with enhanced host immunity. Using NanoString nCounter gene expression assays, we assessed the expression of genes associated with anti-tumor immune responses. The transcriptomic changes in H358 tumors receiving scL-RB94 versus the untreated baseline are illustrated in a volcano plot to show differential gene expression pattern (Figure 3A). We applied the criteria of change greater than 50% and a false discovery rate (FDR) < 0.05 to define differentially expressed genes. As a result, eight upregulated genes were found to be differentially expressed between the tumors receiving scL-RB94 and the untreated baseline (Figure 3A). Genes upregulated in the scL-RB94 treatment group included those related to antigen presentation and innate immune responses (HLA-DRA, A2M, HLA-DQA1, CCL21, and PIK3CG) and complement responses (C5AR1). In addition, cAMP-dependent transcription factor 3 (ATF3) inhibits tumor cell proliferation by inducing tumor cell apoptosis [23] and functions as an immunomodulator by interacting with nuclear factor κB (NF-κB) and repressing proinflammatory cytokines [24,25]. Interestingly, it is believed that cancer/testis antigen 1 encoded by the CTAG1B is an immunogenic protein inducing spontaneous cellular and humoral immune responses in patients with NSCLC [26]. To further understand the molecular and biological pathways that are enriched in tumors receiving scL-RB94 treatment, we performed gene ontology (GO) term and pathway analyses using the ToppGene Suite online gene set enrichment analysis tool (https://toppgene.cchmc.org/enrichment.jsp, accessed on 9 June 2022), which revealed several enriched pathways (Figure 3B). Enriched GO molecular functions included MHC class II receptor binding and complement component binding. Enriched GO biological processes included innate immune response, positive regulation of immune system processes, and activation of immune response. Enriched GO cellular components included MHC protein complex. Enriched pathways included those involved in T-cell activation, allograft rejection, and inflammation mediated by chemokine and cytokine signaling (Figure 3B). These data indicate that scL-RB94 treatment can improve immune responses affecting various pathways of anti-tumor immunity in the host.

### 3.5. scL-RB94 Inhibits Growth of NSCLC Tumor via NK Cells

Because the cytotoxicity of RB94 is restricted to human cancer cells, we employed xenograft tumor models in athymic nude mice to allow human tumors to be established. However, athymic nude mice lack T cells, which consequently narrowed our focus to innate immune responses. Interestingly, following the exposure of NSCLC cells to scL-RB94 in culture, we observed a time-dependent increase of ligands (ULBP2, MICA, and MICB) for NK cell activation receptors at the mRNA level compared with untreated H292 and H358 cells (Figure 4A and Appendix A). FACS analysis of tumors from mice treated with scL-RB94 showed similar results at protein level (Figure 4B). After receiving scL-RB94 treatment via tail-vein injection, H358 tumors displayed a large increase in surface expression of ULBP1 and ULBP2 compared with untreated tumors or tumors treated with the scL-vec control. These results indicate that scL-RB94 treatment could increase expression of ligands for NK cell activation receptors, thereby priming tumor cells to stimulate NK cells. Importantly, FACS analyses revealed an increase of tumor-infiltrating NK cells following treatment with scL-RB94 in H292 and H358 tumor models (Figure 4C,D). However, there was no increased tumor-infiltration of NK cells in mice treated with scL-vec compared with untreated mice (Figure 4D). Overall, upregulation of NK-cell-activation ligands and increased tumor-infiltration of NK cells suggest that NK cells could be the major contributor to the observed anti-tumor immunity with scL-RB94 treatment in our NSCLC models. To examine the effect of NK cell depletion on anti-tumor effects of scL-RB94 treatment, we treated mice bearing H358 tumors with anti-NK1.1 antibodies prior to scL-RB94 treatment [27]. Although we did not observe complete depletion of NK cells, the antibody treatment significantly lowered the tumor-infiltrating NK cells to levels similar to those of untreated tumors (Figure 4D). Strikingly, inhibition of tumor growth previously seen with scL-RB94 treatment was nearly absent in the NK-cell-depleted mice (Figure 4E), suggesting that anti-tumor activity of scL-RB94 might be mainly facilitated by NK-cell-mediated innate immune responses. Importantly, we observed a significant upregulation of both human and mouse IL-15 and CXCL1 in H358 tumors after scL-RB94 treatment, assessed by RT-qPCR (Figure 4F). IL-15 is a potent immune-stimulating cytokine that promotes the survival, proliferation, and cytolytic capacity of NK cells to potentiate NK-cell-mediated immune responses [28,29]. CXCL1 has been shown to promote cytokine secretion and degranulation by NK cells [30]. Deletion of CXCL1 reduced the ability of NK cells to clear tumors, by downregulating CD107a and IFNg [30]. Taken together, these results suggest that scL-RB94 treatment could augment anti-tumor host immune responses mediated by NK cells.

### 3.6. scL-RB94 Treatment Reduces Immunosuppressive M2 Macrophages

Macrophages play an important role in anti-tumor immunity, both inhibiting (anti-tumoral M1 macrophages) and promoting (pro-tumoral M2 macrophages) tumor growth. Thus, we examined the effects of scL-RB94 on tumor-associated macrophage phenotypes. Treatment of tumor-bearing mice with scL-RB94 appeared to modulate the polarization of macrophages. Flow cytometry analysis of H358 tumors revealed that scL-RB94 treatment significantly reduced the number of pro-tumoral CD206+ M2 macrophages compared with untreated tumors, while M1 macrophages remained unchanged (Figure 5A). As a result, scL-RB94 treatment significantly increased the M1/M2 ratio, supporting an anti-tumoral TME. In addition, IHC staining of Arg1, a marker for M2 macrophages, was significantly decreased in H358 and H292 tumors after scL-RB94 treatment (Figure 5B). Staining for iNOS (a marker for M1 macrophages) was not significantly changed by scL-RB94 treatment. Thus, our results indicate that scL-RB94 treatment alters the phenotype of tumor-associated macrophages, and this effect may decrease the tumor’s ability to evade the immune system.

## 4. Discussion

Despite recent therapeutic advancements (e.g., molecularly targeted therapeutics and immune checkpoint inhibitors), the therapeutic efficacy of novel agents has been limited to only small percentages of patients who harbor specific biomarkers or genetic alterations, leaving the majority of NSCLC patients with poor prognosis and a high mortality rate [31,32,33,34]. Hence, novel therapeutic approaches are urgently needed to extend clinical benefits to a broader patient population and improve patient outcomes in advanced NSCLC.

A loss of tumor-suppressor function predisposes cells to malignant transformation, and disruption of tumor-suppressor gene pathways is a hallmark of neoplastic cells including lung cancer. Thus, uses of tumor suppressor genes as anti-cancer therapeutics have been investigated in preclinical and clinical settings. However, the lack of a systemic delivery method represents a fundamental barrier to the use of tumor-suppressor gene-therapy agents in the context of disseminated human cancers [19]. In this context, our tumor-targeted nanomedicine scL-RB94, which systemically delivers the tumor-suppressor RB94 gene selectively to tumor cells, offers a very promising therapeutic option to improve the poor outcomes currently seen among advanced NSCLC patients. Importantly, in a previous phase I clinical trial for genitourinary cancers, scL-RB94 (SGT-94) nanocomplex demonstrated its safety and clinical activity in patients [19]. Notably, biopsies were taken from metastatic lesions in the lung and from the adjacent normal lung tissue, and the exogenous RB94 protein was only found in the metastatic lesions and not in the normal tissue, demonstrating excellent tumor-targeting specificity of SGT-94 [19]. However, more research is required to elucidate the mechanism behind the efficacy of this novel nanomedicine. In the current study, we further tested the tumor-suppressive efficacy of scL-RB94 and investigated its anti-tumor mechanism in preclinical models of NSCLC.

A recent clinical study with a cohort of advanced NSCLC patients (with stage III and IV disease) showed that RB mutant status was strongly associated with worse outcomes in patients, with median overall survival of 8.3 months for patients with mutant RB compared to 28.3 months for patients with wild-type RB [35]. The frequency of RB mutation was relatively low, affecting 8.2% of these patients. Interestingly, RB94 exhibited anti-tumor efficacy in RB-negative and RB-positive human tumors, while the tumor-suppressive effect of full-length RB protein was limited to RB-negative tumors [10,36]. These results are in line with our observation that RB94 was cytotoxic to multiple NSCLC types irrespective of their genetic makeup (i.e., RB status), suggesting that SGT-94 could be expanded to other tumor types irrespective of their native RB status [19]. Although our observations require further research to reveal the mechanism(s) underlying anti-tumor efficacy independent of RB status, the current results suggests that scL-RB94 nanomedicine has potential to benefit a broader patient population, i.e., not requiring specific biomarkers or genetic alterations.

In our xenograft models (athymic mice bearing human NSCLCs), intravenously administered scL-RB94 nanocomplex was highly effective in suppressing tumor growth in vivo due to a multimodal mechanism, acting directly on the tumor cells (i.e., anti-proliferation, induction of apoptosis, and increasing immunogenicity of the tumor) and modifying the immune environment in the tumor tissue (i.e., reducing immune suppression and enhancing anti-tumoral innate immune responses).

The canonical role of RB protein as a tumor suppressor Is related to its inhibition of cell-cycle progression [11]. In addition, RB is also involved in regulating DNA damage responses, inducing senescence, and activating apoptosis [14,37]. Our results suggest that the RB variant RB94 could restore or enhance the canonical RB functions, to decrease proliferation and enhance apoptosis of NSCLC cells, although direct comparisons between truncated RB94 and full-length RB were not made in the present study. We observed that scL-RB94 treatment effectively triggered apoptosis in different human NSCLC cell lines irrespective of their RB status and was able to substantially suppress tumor growth in vivo.

Importantly, the RB pathway also has been implicated in the regulation of innate immune functions [37,38]. A recent study showed that miR-181a-mediated inhibition of RB, leading to oncogenic transformation of fallopian tube secretory epithelial cells through the regulation of innate immune signaling, along with co-inhibition of stimulator-of-interferon-genes (STING), suggesting a function for RB in immune regulation [37]. In another study, disruption of the RB pathway in liver cells resulted in increased cellular proliferation as well as decreased immune function, with a similar gene signature to that associated with hepatocellular carcinoma [38]. In conjunction with these reports, we observed that expression of exogenous RB94 via scL-RB94 treatment resulted in decreased cellular proliferation as well as increased immune function in NSCLC.

In our study, induction of RB94 expression increased immune functions as part of a multifaceted process. Treatment with scL-RB94 induced immunogenic changes of NSCLC by triggering immunogenic cell death (ICD), increasing type I interferon responses, and increasing expression of immune recognition molecules (e.g., ULBP2, MICA, and MICB for NK cells) and antigen processing and presentation machinery including peptide loading molecules (TAP1/2) and the antigen presentation molecule (HLA-ABC). Of significance was the increased infiltration of NK cells in tumors of mice receiving scL-RB94 treatment, while an antibody-mediated depletion of NK cells blunted the anti-tumor effect of scL-RB94, indicating an important contribution of NK cells to the observed anti-tumor activity of RB94 gene therapy. scL-RB94 treatment also increased the secretion of immune-stimulating cytokines IL-15 and CXCL1 that promote the survival, proliferation, and cytolytic capacity of NK cells. Furthermore, scL-RB94 treatment decreased immunosuppressive M2 macrophages in tumors, lowering the ability of tumors to evade the host immune system. When these results are viewed together, they indicate that scL-RB94 not only inhibits tumor growth via apoptosis and/or cell-cycle arrest, but also acts to alter tumors in a way that augments or enhances host immune responses.

It is now well accepted that gene therapy as a single agent might be far less effective than when used in combination with conventional therapeutic modalities (e.g., chemotherapeutic agents or ionizing radiation) [15]. Our previous study showed that adding scL-RB94 to gemcitabine significantly enhanced therapeutic modality for the treatment of human bladder carcinoma [15]. In other studies, adenovirus-mediated RB94 gene therapy enhanced the efficacy of radiation treatment of human head and neck squamous cell carcinoma [14] or esophageal squamous cell carcinoma [39], resulting in significant tumor regression in vivo. Our data suggest that scL-RB94 might be effective when combined with immune checkpoint inhibition (e.g., anti-PD-1), considering its ability to enhance antigen processing and presentation and to lower immunosuppression. Thus, we believe that scL-RB94 may be a good candidate to augment the current first-line treatment of patients with advanced NSCLCs, and further evaluation is warranted.

## 5. Conclusions

In summary, we report that truncated tumor suppressor RB94 gene therapy via tumor-targeted nanocomplex (scL-RB94) was able to mediate anti-tumor activity in animal models of human NSCLC, by not only increasing tumor cell death but also increasing immunity of tumor cells and alleviating immunosuppression. These findings suggest that scL-RB94 has potential as a novel therapeutic entity capable of improving outcomes in the treatment of advanced NSCLC. Our observations provide motivation to evaluate scL-RB94 in patients with advanced NSCLC.

## Figures and Tables

**Figure 1 cancers-14-05092-f001:**
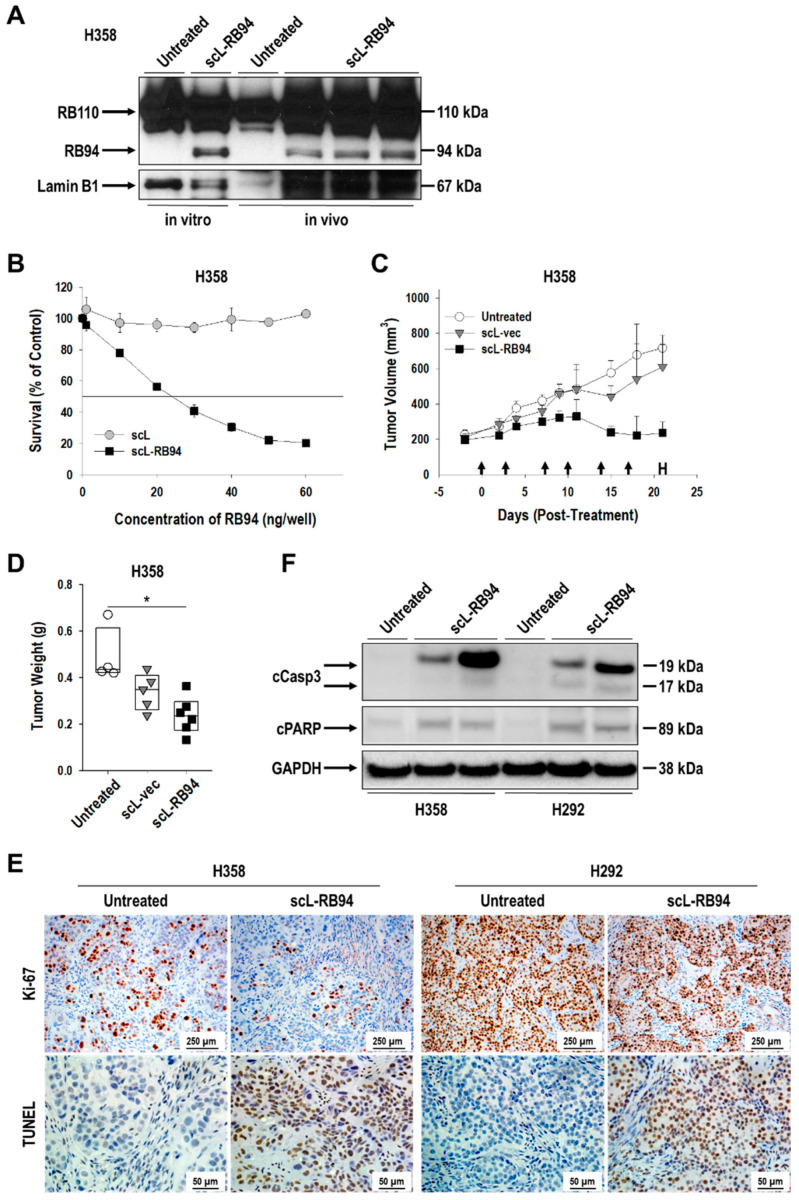
Expression of exogenous RB94 via scL-RB94 nanocomplex inhibits the proliferation and survival of human NSCLCs. (**A**) Western blot analysis depicting the expression of exogenous RB94 protein (94 kDa) in H358 cells both in vitro and in vivo after treatment with scL-RB94 nanocomplex. The expression of endogenous RB protein (110 kDa) is also shown. Expression of lamin B1 protein was utilized as an internal control for protein loading. (**B**) Cell viability in H358 cells was measured by XTT assay 72 h after transfection with increasing concentrations of scL-RB94 or empty nanocomplex without payload (scL). For in vivo study, athymic mice bearing subcutaneous H358 tumor xenograft were treated with either scL-vec or scL-RB94 (30 µg/injection/mouse), as indicated by arrows in (**C**). (**C**) Tumor growth was monitored by measuring tumor volumes (*N* = 5–8). (**D**) Tumors were harvested and weighed on day 21 (indicated by H). * *p* < 0.05, one-way ANOVA. In another in vivo study, athymic mice bearing either H358 or H292 tumors were treated with scL-RB94 (30 µg/injection/mouse, two injections over 3 days). Forty-eight hours later, harvested tumors were processed for IHC and Western blot analyses (*N* = 8). (**E**) Representative staining for the proliferation marker Ki-67 (upper panels) and the apoptosis marker TUNEL (lower panels). (**F**) Western blot analysis assessing changes in cleaved caspase-3 (cCasp3) and cleaved PARP (cPARP) protein levels. Expression of GAPDH protein was utilized as an internal control for protein loading. Original complete Western blot images for Figure 1 are provided in Appendix A.

**Figure 2 cancers-14-05092-f002:**
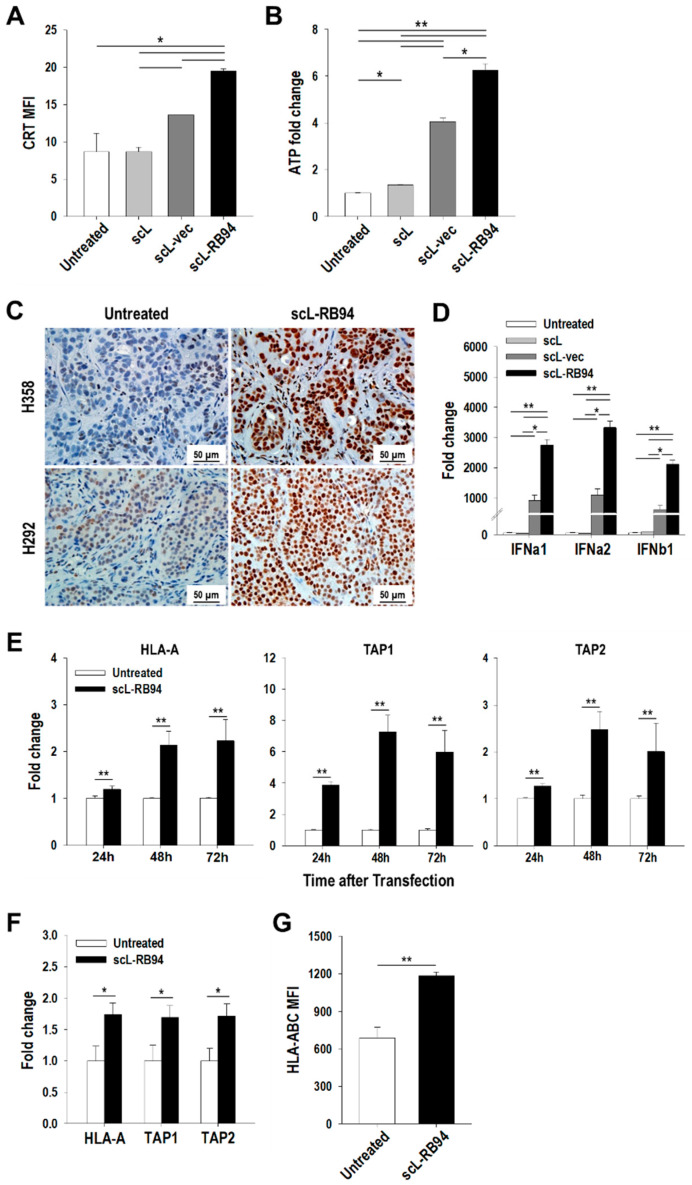
scL-RB94 increases immunogenicity of NSCLC cells. (**A**) Flow cytometry analysis of surface expression of CRT and (**B**) ATP concentration in cell culture supernatants in H358 cells after transfection with scL-RB94 nanocomplex in vitro. In the control, H358 cells were transfected with either scL-vec nanocomplex carrying the control plasmid vector without RB94 or empty scL nanocomplex without the payload. * *p* < 0.05, ** *p* < 0.001, one-way ANOVA. (**C**) Representative HMGB1 staining of H358 and H292 tumors 48 h after scL-RB94 treatments (30 µg/injection/mouse, two injections over 3 days, *N* = 8). Quantitative RT-PCR analysis of (**D**) type I interferons and (**E**,**F**) antigen presentation and processing molecules, HLA-A, TAP1, and TAP2 in (**D**,**E**) H358 cells in vitro and (**F**) H358 tumors in vivo after treatment with scL-RB94. * *p* < 0.05, ** *p* < 0.001, one-way ANOVA or Student’s *t* test. (**G**) Flow cytometry analysis of expression of HLA-ABC in H358 tumors after treatment with scL-RB94 in vivo (*N* = 5–8). ** *p* < 0.001, Student’s *t* test.

**Figure 3 cancers-14-05092-f003:**
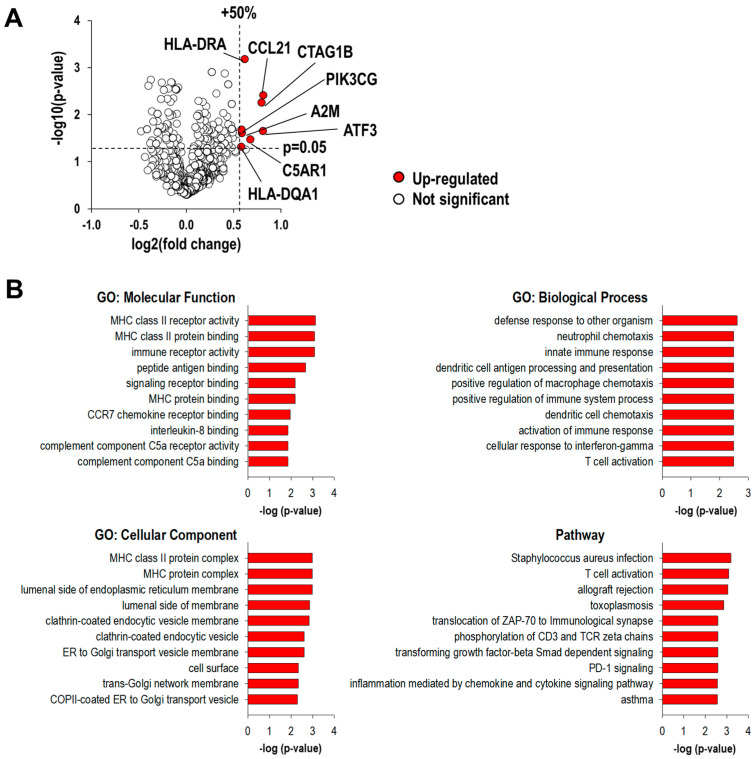
Transcriptomic analyses of H358 tumors using nCounter PanCancer Human IO 360 assay. (**A**) Volcano plot showing differential expression of genes in tumors treated with scL-RB94 versus the baseline of untreated tumors. The volcano plot is presented as fold change in gene expression [log_2_(fold change)] against statistical significance of change [−log_10_(*p*-value)]. (**B**) ToppGene Suite analysis of differentially expressed genes. Groups reflect main categories of gene ontology (GO) terms: molecular function, biological process, cellular component, and canonical pathway. Vertical and horizontal axes represent each category term and −log_10_(FDR adjusted *p*-value) of the corresponding term, respectively.

**Figure 4 cancers-14-05092-f004:**
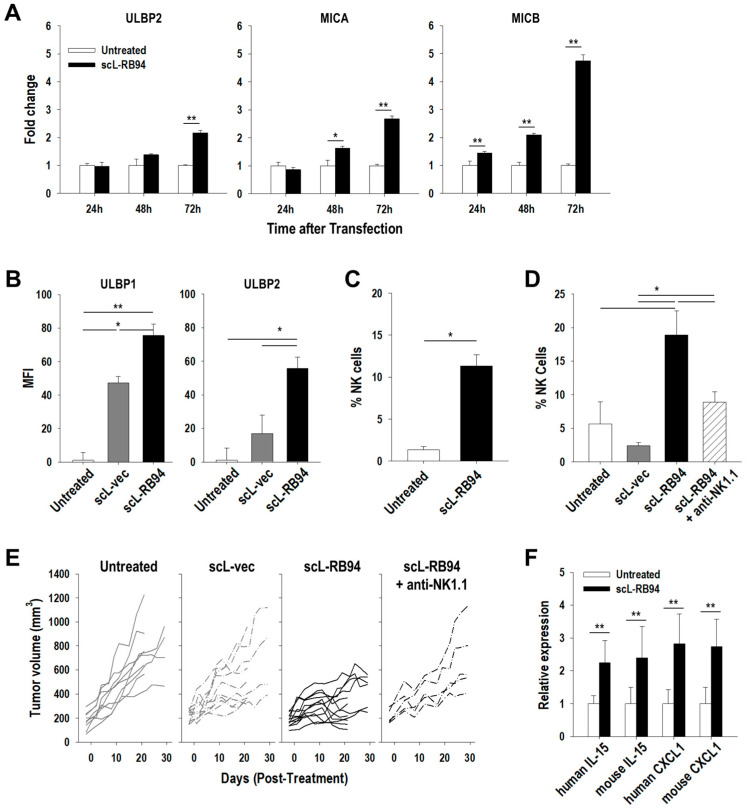
scL-RB94 inhibits growth of NSCLC tumors via NK cells. (**A**) Quantitative RT-PCR analysis of ligands associated with NK cell activation (ULPB2, MICA, and MICB) in H292 cells after treatment with scL-RB94 in vitro (*N* = 6). * *p* < 0.05, ** *p* < 0.001, Student’s *t* test. Flow cytometry analysis of (**B**) ligands associated with NK cell activation (ULBP1 and ULBP2) in H358 tumors and (**C**) NK cells infiltrating H292 tumors after scL-RB94 treatment (30 µg/injection/mouse, two injections over 3 days). NK cells were identified by gating CD45^+^ live cells with NK1.1^+^NKp46^+^. * *p* < 0.05, ** *p* < 0.001, one-way ANOVA or Student’s *t* test. In another procedure, athymic mice with H358 tumors were treated with either scL-RB94 or scL-vec (30 µg DNA/injection/mouse, total eight injections over 4 weeks). In one cohort of scL-RB94 treated mice, the NK cells were depleted with anti-NK1.1 antibody, 24 h prior to scL-RB94 treatment (*N* = 5–8). (**D**) Flow cytometry analysis of NK cells infiltrating H358 tumors. * *p* < 0.05, one-way ANOVA. (**E**) Tumor volume for individual mice was measured and graphed for each group. (**F**) Quantitative RT-PCR analysis of cytokines (IL-15 and CXCL1) produced by the H358 tumor (human) and host (mouse). ** *p* < 0.001, Student’s *t* test.

**Figure 5 cancers-14-05092-f005:**
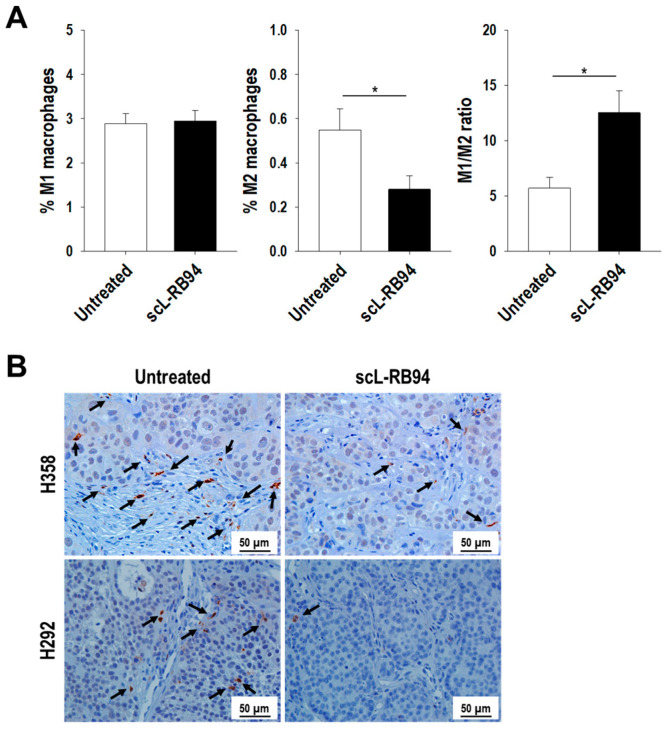
scL-RB94 reduces immunosuppressive M2 macrophages. Athymic mice bearing either H358 or H292 tumors were treated with scL-RB94 (30 µg/injection/mouse, two injections over 3 days). Forty-eight hours later, harvested tumors were processed for flow cytometry or IHC analyses (*N* = 8). (**A**) Macrophages infiltrating H358 tumors were identified by flow cytometry using gates for M1 macrophages (CD11b+F4/80+CD86+) and M2 macrophages (CD11b+F4/80+CD206+) in CD45^+^ live cells. * *p* < 0.05, Student’s *t* test. (**B**) Representative staining for Arg-1, a common marker for M2 macrophages in H358 and H292 tumors. Arrows indicate Arg-1 positive cells.

## Data Availability

Not applicable.

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
