# Peer review of "Nanomedicine-Based Gene Delivery for a Truncated Tumor Suppressor RB94 Promotes Lung Cancer Immunity"

_cancers, 2022, doi:10.3390/cancers14205092_

Round 1

Reviewer 1 Report

This manuscript describes the past and current knowledge on anti-tumor potency of nano medicine-based gene delivery for tumor suppressor RB94. In addition, author summarized potential of scL-RB94 to improve outcomes in lung cancer patients It is informative and well-written research paper.

My only concern is that figure legends and titles are not clearly labeled and current references must be cited.

I would like to recommend this article to be published in cancers based on author's careful revision on references and figures.

Reviewer 2 Report

Review of “Nanomedicine-based gene delivery for a truncated tumor suppressor RB94 promotes lung cancer immunity” by Kim et al.

This study provided interesting observations in the anti-tumor function of nanomedicine scL-RB94 and would be interested by many. Overall, it is well written and the analysis are rigorous. I would recommend this manuscript for publish. 

One remaining question is that whether scL-RB94 has a physiological impact on the control animals? I saw that human cells are not affected by RB94 but not sure for animals. It would be good to mention or discuss this in the manuscript.

Minor comment: Few statistical analyses are missing in the figures, e.g., Figure 2D, Figure 4BD.

Reviewer 3 Report

It was a manuscript about the potential application of  scL-RB94 for the treatment of lung cancer. Here are some comments on this sudy that should be considered before publication:

1-      Please introduce all the abbreviations at their first usage.

2-      Please separate in vitro tests from the in vivo ones in the method part.

3-      DLS is not a sufficient test for the evaluation of nanomaterials. You need to add other types of characterization tests for that.

4-      Please add the XTT results of other cell lines (A549, H596, and H23), as well.

5-      Why the volume of the H358 tumor didn’t change after 10 days of treatment with scL-RB94, while the tumor weight was decreased?

6-      Please use more updated references.

Round 2

Reviewer 3 Report

Thanks for addressing the comments.